# Effects of *Bifidobacterium longum* and *Lactobacillus rhamnosus* on Gut Microbiota in Patients with Lactose Intolerance and Persisting Functional Gastrointestinal Symptoms: A Randomised, Double-Blind, Cross-Over Study

**DOI:** 10.3390/nu11040886

**Published:** 2019-04-19

**Authors:** Paola Vitellio, Giuseppe Celano, Leonilde Bonfrate, Marco Gobbetti, Piero Portincasa, Maria De Angelis

**Affiliations:** 1Department of Soil, Plant and Food Sciences, University of Bari Aldo Moro, via Amendola 165/a, 70126 Bari, Italy; paolavitellio91@gmail.com (P.V.); g.celano1@gmail.com (G.C.); 2Clinica Medica “A. Murri”, Department of Biomedical Sciences and Human Oncology, University of Bari Medical School, 70121 Bari, Italy; leonildebnf@gmail.com; 3Faculty of Science and Technology, Free University of Bolzano, piazza Università, 5, 39100 Bolzano, Italy; marco.gobbetti@unibz.it

**Keywords:** lactose intolerance, probiotics, vitamin B6, microbiome, metabolome

## Abstract

Functional gastrointestinal symptoms are frequent, and may be driven by several pathogenic mechanisms. Symptoms may persist in lactose intolerant (LI) patients (i.e., subjects with intestinal lactase deficiency, lactose malabsorption producing symptoms), after a lactose-free diet. Our hypothesis was that probiotic and vitamin B6 treatment may be useful to alleviate symptoms in LI patients through a positive modulation of gut microbial composition and relative metabolism. We aimed to test the efficacy of a novel formulation of *Bifidobacterium longum* BB536 and *Lactobacillus rhamnosus* HN001 plus vitamin B6 (ZR) in 23 LI subjects with persistent symptoms during a lactose-free diet. Symptoms, microbiome, and metabolome were measured at baseline and after 30 days in a crossover, randomized, double-blind study of ZR versus placebo (PL). Compared with PL, the administration of probiotics and vitamin B6 significantly decreased bloating (*p* = 0.028) and ameliorated constipation (*p* = 0.045). Fecal microbiome differed between ZR and PL. ZR drove the enrichment of several genera involved in lactose digestion including *Bifidobacerium*. Moreover, the relative abundance of acetic acid, 2-methyl-propanoic acid, nonenal, and indolizine 3-methyl increased, while phenol decreased. Our findings highlight the importance of selected probiotics and vitamin B6 to alleviate symptoms and gut dysbiosis in lactose intolerant patients with persistent functional gastrointestinal symptoms.

## 1. Introduction

Functional gastrointestinal diseases (FGIDs) are the most common cause of gastrointestinal disturbance in global population. Once organic (i.e., inflammatory or neoplastic) causes are ruled out, symptoms include a wide range of disorders [1] affecting the esophagus or upper gastrointestinal tract (i.e., functional dyspepsia, postprandial distress syndrome, or epigastric pain), bowel disorders (i.e., irritable bowel syndrome (IBS), constipation, diarrhea, or bloating), biliary and anorectal disorders, or dyspepsia. Several factors are involved in the genesis of FGIDs, and include intestinal dysbiosis [2,3], genetic predisposition, perception, diets, and intestinal dysmotility [4]. The exact physiopathology of FGIDs, however, remains poorly elucidated so far. FGIDs can also coexist with types of food intolerance. Lactose intolerance (LI) (due to lactase deficiency, down-regulation of lactase activity, lactose malabsorption causing abdominal symptoms upon lactose-containing products) [5,6,7] and fructose intolerance (due to fructose malabsorption with symptoms upon ingestion of fructose-containing foods) in adults [8] can easily mimic several FGIDs. Self-reported perception of LI can also affect the behaviour of patients in terms of diagnosis and therapeutic approaches [9,10,11]. However, even with a lactose- or fructose-restricted diet regimen, functional symptoms like constipation or diarrhoea, functional bloating, or dyspepsia may persist.

Recently, the idea of improving gut microbiota composition with some selected probiotic strains represents a valid therapeutical approach in FGIDs, even with concomitant LI [12]. Probiotics are defined as live bacteria or yeasts that positively supplement the gut microbiota [13]. Probiotic supplementation is a valid approach to maintain the balance of the intestinal microbiota, the composition of which could be altered by several factors (stress, lifestyle, diet, antibiotic therapies, and so on) [14]. Hence, probiotics can be administered for treatment of FGIDs, as well as carbohydrates malasorbition [15].

In particular, lactic acid bacteria (LAB) constitute a large part of intestinal microbiota and their probiotical properities are well documented [16]. The genus *Lactobacillus* is a source of interest in the prevention of some diseases including LI [17]. Moreover, the beneficial use of LAB and probiotics in general aids in the extraction of energy and nutrients, such as short-chain fatty acids (SCFAs) (acetate, propionate, butyrate) and amino acids from food [18]. Furthermore, the production of SCFAs stimulates ileal propulsive contractions, releases neuroendocrine factors, acidificates the intracolonic pH, and relaxes intestinal gut tissues [19].

In addition, there is evidence that low intake of vitamin B6 might be associated with FGIDs such as IBS [20]. Notably, B6 is shown to have pleiotropic functions for human health [21].

The present study aimed to investigate the effects of a novel formulation of *Bifidobacterium longum* BB536 and *Lactobacillus rhamnosus* HN001 with vitamin B6 on symptoms, gut microbiota, and matabolome in a cohort of patients with persisting FGIDs on a lactose-free diet because of a prior diagnosis of LI.

## 2. Materials and Methods

### 2.1. Patients and Study Design

The initial group consisted of 135 adult symptomatic patients of both genders (54 males and 81 females). Figure 1 depicts the consort flow-chart of screened patients. Exclusion criteria were organic diseases with a diagnosis of structural abnormality of the gastrointestinal tract (i.e., inflammatory bowel diseases such as Crohn’s disease, ulcerative colitis), pregnancy, abdominal surgery within the previous six months, infective diseases, drug or alcohol abuse, metabolic disturbances, mental illness, concomitant immunological, haematological or neoplastic disease, severe hepatic insufficiency (i.e., Child–Pugh class A–C), severe heart failure (NYHA class III–IV), and inability to provide the informed consent. Also excluded were 15 patients with a diagnosis of IBS according to the Rome IV criteria. IBS is characterized by recurrent chronic abdominal pain or discomfort and changes in stool or improvement with defecation, in the absence of detectable organic causes [22,23]. The 99 remaining patients underwent the hydrogen (H2) breath test to detect LI intolerance and 75 patients (19 males and 56 females, mean age 46 ± 3.1 SE years, range 20–67) were positive (Figure 1). The patients were encouraged to introduce lactose-free dairy products to undergo a lactose-free diet for at least six months. Patients (*N* = 34) were ultimately enrolled if symptoms (i.e., altered bowel habits such as constipation or diarrhea, bloating, and abdominal pain) persisted after six months. Because of 11 drop-out patients, the final cohort consisted of 23 subjects.

### 2.2. Lactose Breath Test

The lactose hydrogen breath test (LH2-BT) was performed according to previous guidelines [23] using lactose as fermentable substrate (500 mL of cow’s fresh whole milk containing 25 g of lactose). To avoid interferences with the fermentation process, a carbohydrate-free diet was prescribed the day before the test [24]. Smoking, physical exercise, and eating were not permitted 2 h before and during all test (180 min) [25,26]. To measure the time-dependent concentrations of H2 in breath samples (as ppm), the operator used an automatic portable analyzer (Gastro + Gastrolyzer^®^, Kent, England), with an accuracy of ±2 parts per million (ppm), a resolution of 1 ppm, and a range of 0–500 ppm. Subjects had to collect a breath sample across a mouthpiece connected to the Gastrolyzer at baseline and every 30 minutes. H2 levels were reported together with the presence of gastrointestinal symptoms (bloating, abdominal pain) and bowel movements [7]. The cut-off value for defining lactose malabsorption (absence of gastrointestinal symptoms) or LI (malabsorption and symptoms) was an H2 concentration of at least 20 ppm above the baseline (usually 1–2 ppm). Thus, only patients with LI (but not with simple lactose malabsorption or normal at H2 breath test) were enrolled.

### 2.3. Randomization and Masking

The study was a crossover, randomized, double-blind, placebo-controlled study (Figure 2), and was carried out at the Clinica Medica “A. Murri”, Department of Biomedical Sciences & Human Oncology, University of Bari Aldo Moro between January 2017 and December 2018. An independent researcher performed the randomization using a computer-generated randomization list. Alfasigma S.p.A. (Milano, Italy) provided the packets containing Zircombi^®^ (ZR) and placebo (PL), and the randomization sequence for every patient according to the intervention treatments (ZR and PL). ZR and PL products were indistinguishable in appearance.

### 2.4. Double Blind Cross Over Challenge

During the run-in (Ri) period, subjects fitting the inclusion criteria were randomized to receive either probiotic (ZR: 3 g as packets containing *B. longum* BB536 four billion CFU, *L. rhamnosus* HN001 one billion CFU with B6 vitamin 1.4 mg) or placebo (PL: similar packets containing maltodextrins, corn starch, silicon dioxide, and no probiotic).

The study included a first period of one month of treatment followed by 15 days of wash-out, and a second period of one month of treatment after cross-over. Nonsteroidal anti-inflammatory drugs, anticoagulant, antibiotics, or probiotics were prohibited during the two weeks before baseline and throughout the study. Other treatments (i.e., antispasmodic, triptans, anticholinergics, motility regulating drugs and osmotic laxative, and antidepressant or anxiolytic drugs) were also prohibited at stable doses in the four weeks before randomization and during the whole study. Analyses were performed at days 0 (T0), 30 (T30), 45 (T45), and 75 (T75), and included symptoms, antrophometric evaluations, fecal microbiota, and related metabolome measurements.

In order to assess compliance to study treatment and to record adverse events, patients were interviewed on a regular basis by medical personnel blinded to the regimen; compliance was calculated as the percentage of returned study product and a compliance was considered acceptable if >80%.

The study adhered to the Declaration of Helsinki and was approved by the Ethical Institutional Review Board of Institutional Ethics Committee of Bari University Hospital (study number 4651 ref. 11061CE, 02-10-2017). Written, informed consent was obtained from the patients, who were fully informed of the nature and purpose of the study. The study was registered in the Protocol Registration System Clinical Trial.gov (ClinicalTrials.gov Identifier: NCT03815617).

### 2.5. Questionnaires

Patients were evaluated for intensity of symptoms (abdominal pain and bloating, measured as visual analogue scale (VAS) in mm ranging from 0 to 100) and bowel movements were by the use of Bristol stool form scale (BSFS). Lifestyle and daily intake of foods were assessed using the MEDSTYLE questionnaire [8]. The adherence to a Mediterranean diet was calculated according to Sofi et al. [27].

### 2.6. Outcomes

The primary outcome was to determine whether dietary supplementation with ZL as compared with PL was able to improve symptoms (bloating and abdominal pain), as assayed by visual analogue scale (VAS). The scale is displayed as a horizontal line ranging from 0 to 100 mm, where 0 is “no perception” and 100 is “the worst possible perception” [28].

The secondary outcome was the assessment of changes of bowel movements measured by the Bristol stool form scale. This scale represents a diagnosis instrument for classifying stool (1–7) and thus bowel habits. In particular, type 1–2 indicate constipation, type 3–4 indicate normal stool, and type 5–7 indicate diarrhoea [29]. A symptomatic response characterized by a decrease of at least 30% of the global VAS or Bristol stool form scale from the PL and ZR was defined as a clinical response.

### 2.7. Fecal Collection

Each subject provided two fecal samples at each study time point. After collection, feces were mixed with RNA later (Sigma-Aldrich, St. Louis, MO, USA) (ca. 5 g, 1:2 *w/v*) or with Amies transport medium (Oxoid LTD, Basingstoke, Hampshire, England) (ca. 15 g, 1:1 *w/w*), as previously described [30]. Fecal samples suspended in RNA later were stored at −80 °C for RNA extraction. Samples diluted with Amies Transport were used for plate count (supplementary materials) and metabolome analysis.

### 2.8. RNA Extraction and Fecal Microbiome

Total genomic bacterial RNA was isolated from frozen stool samples using the Stool Total RNA purification KIT (Norgen Biotek Corp., Ontario, Canada), according to the manufacturer’s instructions. The quality and concentration of the RNA were determined by spectrophotometric measurements at 260, 280, and 230 nm through the NanoDrop ND-1000 spectrophotometer. Total RNA extracted (approximately 2.5 μg) was retrotranscribed in cDNA using random hexamers and the Tetro cDNA synthesis kit (Bioline USA, Inc., Taunton, MA, USA), according to the manufacturer’s instructions.

Bacterial Microbiome was estimated by 16S rRNA. A 16S metagenetic analysis was carried out at Genomix4life (spin-off of the University of Salerno, Fisciano, Italy) using the Illumina MiSeq platform. The V3–V4 region of the 16S rRNA gene was amplified for analysis of diversity inside the domains of bacteria [31]. PCR and sequencing analyses, quality control, and taxonomic assignment were carried out according to the protocol of Genomix4life. Shannon diversity and alpha diversity indices were calculated using Qiime analysis [32].

### 2.9. Fecal Metabolome

Fecal samples, placed into 10 mL glass vials, were sealed with polytetrafluoroethylene (PTFE)-coated silicone rubber septa, and then equilibrated for 10 min at 40 °C. At the end of sample equilibration, a conditioned 50/30 µm DVB/CAR/PDMS fibre (Supelco, Bellefonte, PA, USA) was exposed to headspace for 40 min to extract volatile compounds by CombiPAL system injector autosampler (CTC Analytics, Zwingen, Switzerland). Volatile organic compounds (VOCs) were thermally desorbed by immediately transferring the fibre into the heated injection port (220 °C) of a Clarus 680 (Perkin Elmer, Beaconsfield UK) gas chromatography equipped with an Rtx-WAX column (30m × 0.25mm i.d., 0.25 μm film thickness) (Restek) and coupled to a Clarus SQ8MS (Perkin Elmer) with source and transfer line temperatures kept at 250 and 210 °C, respectively. The injection was carried out in splitless mode, and helium was used as the carrier gas at flow rate of 1 mL/min. The oven temperature was initially set at 35 °C for 8 min, then increased to 60 °C at 4 °C/min, to 160 °C at 6 °C/min, and finally to 200 °C at 20 °C/min and held for 15 min. Electron ionization masses were recorded at 70 eV in the mass-to-charge ratio interval, which was m/z 34 to 350. The GC-MS generated a chromatogram with peaks representing individual compounds. Each chromatogram was analysed for peak identification using the National Institute of Standard and Technology 2008 (NIST) library. A peak area threshold of >1 000 000 and 85% or greater probability of match was used for VOC identification, followed by manual visual inspection of the fragment patterns when required. 4-Methyl-2-pentanol (final concentration 33 mg/L) was used as an internal standard in all analyses, in order to quantify the identified compounds by interpolation of the relative areas versus the internal standard area.

### 2.10. Statistical Analysis

The sample size was calculated assuming a 35% difference in response between treatment and placebo. We estimated that 20 patients would be required for the study to have 80% power and an α error of 5%. A per-protocol analysis was applied to the trial. Normally distributed grouped data were expressed as the means and standard deviation (±SD) and compared using paired and unpaired t-tests. Non-parametric grouped data were expressed as the means (95% confidence interval (CI)) and compared using the Mann–Whitney rank sum test (paired) or Wilcoxon´s signed rank test (unpaired). The randomization list was generated using the online resource available at www.randomization.com. Proportionate data were compared using Fisher´s exact test or the χ^2^ test as appropriate. Differences between groups were analyzed using the two-tailed Student’s t-test for independent samples. *p*-value < 0.05 was regarded as significant [33]. The statistical software Statistica for Windows (Statistica 7.0) was used. Metagenomic and metabolomics data were subjected to permutation analysis using PermutMatrix and principal component analysis (PCA) using Statistica 7.0 for Windows.

## 3. Results

### 3.1. Baseline Characteristics

#### 3.1.1. Symptoms

The final cohort consisted of 23 LI patients (4 males and 19 females). The baseline characteristics according to age, gender, body mass index (BMI), symptoms, and bowel habits appear in Table 1. Females constituted 83% of the study group. None of the subjects was obese, as testified by a mean of BMI of 23.2 Kg/m^2^ and median of 22.3 Kg/m^2^. With respect to symptoms, abdominal pain was of intermediate intensity; meanwhile, bloating had a higher score. The bowel habit analysis was in favour of patients suffering from constipation.

#### 3.1.2. Diet

The amounts of daily micro/macronutrients at baseline appear in Appendix A. Notably, patients followed a lactose-free diet for the six months prior to the enrollment and during the whole study period. Dietary fiber intake was lower than the recommended amount of daily fiber (adults = 12.6–16.7 g /1000 kcal, SINU 2014). Also, the intake of daily carbohydrates (41.64% of the total caloric intake) was lower than the recommended guidelines level (45%–60% daily calories intake, SINU 2014). In contrast, the total protein intake was within the reference range of 15%–20%, with a prevalence of animal-type proteins versus plant protein. The total daily amount of lipids was higher than the recommended amount and represented 20%–35% of the total caloric intake. The percentage of saturated fatty acids was higher than the recommended by the guidelines for the prevention of cardiovascular diseases (<10% of the daily caloric intake, SINU 2014). The major source of monounsaturated fatty acids was olive oil, while the daily amount of vitamin B6 was 0.84 ± 0.30 mg per day, which was lower than the recommended daily average requirement (1.1 mg/day). The adherence score to Mediterranean diet for 96% patients was in the range of 10–15 points (sufficiently adherent), while 4% of patients were included in the range of 5–9 points (low adherence).

### 3.2. Clinical Scores during the Double Blind, Cross-Over Challenge

The BMI remained stable throughout the observation period, as well as the intake of micro- and macronutrients. The symptom and bowel habits analysis according to sequence is reported in Appendix A. The treatment with ZR caused a significant (*p* = 0.028) decrease of bloating, compared with placebo (PL) (Figure 3). No effect was evident in the case of abdominal pain (Appendix A).

The globally assessed Bristol score did not differ between placebo and ZR (Appendix A). However, at sub-analysis, the percent of patients with a score within the “normal” range of 3–4 increased from 30.4% to 73.9% after ZR (*p* = 0.0072, chi-squared). In particular, Bristol scores suggestive of constipation (i.e., 1–2) reached a normal range in 7/8 patients (87.5%) after ZR, while in patients with less formed stools (i.e., 5), 3/8 (37.5%) scored normal after ZR (*p* = 0.045). No statistically significant differences were found on the permeability data between placebo and ZR.

### 3.3. Probiotics and Vitamin B6 Affect the Fecal Microbiome of LI Patients

A total number of 1,782,640 raw sequences referred to genus was obtained and analyzed; 1,573,358 reads passed the filters applied through the QIIME split_library.py script. The mean value of the percentage of reads classified to genus was 89%. The average number of species operational taxonomic units (OTUs) identified in the fecal samples of ZR patients (mean value of 233) and PL (mean value of 179) significantly (*p* = 0.049) differed. Shannon index and β-diversity did not differ between ZL and PL samples (data not shown). Compared with 16S-rRNA (total bacteria), the highest significant differences between the fecal microbiota of the two groups (ZR and PL) were found using 16S-rRNA data (metabolically active bacteria). Eleven phyla (Firmicutes, Bacteroidetes, Proteobacteria, Actinobacteria, Chlorobi, Synergistetes, Spirochaetes, Tenericutes, Chloroflexi, Cyanobacteria, Fibrobacteres) were identified (Figure 4). Proteobacteria relative amount was higher (*p* < 0.05) in ZR samples compared with PL samples.

The relative abundance of genus of *Bifidobacterium* was higher (*p* = 0.002) in ZR (0.46%) compared with PL group (0.22%). Compared with PL, ZR drove several genus to a higher amount, including *Slackia*, *Enterococcus,* and *Thricocccus* (*p* < 0.05). *Lactobacillus* genus did not differ between ZR and PL. A decreased relative abundance of *Klesbiella*, *Serratia*, and *Enterobacter* was detected in treated patients compared with PL (*p* < 0.05) (Figure 5). Compared to T0, the treatment ZR increased some components of the culturable microbiota (heterotrophic aerobic and anaerobic bacteria, presumptive lactic acid bacteria and *Bifidobacterium*) (Appendix A).

### 3.4. Probiotics and Vitamin B6 Affect the Fecal Metabolome of LI Patients

We investigated the differences within fecal volatile organic compounds (VOCs) in the ZR group compared with the PL group. VOCs identified from fecal samples (62 compounds) were grouped according to chemical classes, that is, alcohols (12), aldehydes (5), aromatic heterocyclic (2), hydrocarbons (8), ketones (8), short and medium chain fatty acids (SCFA) (10), sulfur compounds (2), esters and methyl esters (1), and terpens (14). The content of the metabolites within the same group largely varied; nevertheless, some significant differences were figured out between the two conditions (Table 2). Compared with PL, some compounds (acetic acid, 2-methyl-propanoic acid, nonenal, and indolizine 3-methyl) were found at significantly (*p* < 0.05) higher levels in the feces of ZR. Within significantly different alcohols (*p* < 0.05), alcohols 1-butanol, phenol, and 1-hexadecanol showed the highest level in the fecal samples of PL. Between the aldehydes compounds, hexanal was found at the highest levels in the feces samples of PL. Terpens were variously distributed within groups, except for limonene, which was significantly higher in the PL group.

### 3.5. Correlations between Microbiome and Metabolome

Several correlations between diet, fecal microbiome, and metabolome existed (Figure 6). In particular, *Bifidobacterium*, *Stachia,* and *Dialister* genera were positively (R > 0.700, FDR < 0.01) correlated with acetic acid, 2-methyl-propanoic acid, D-limonene, and 2-nonanal. *Lactobacillus* genus was positively correlated with acetic acid (R = 0.667, FDR = 0.011). *Bifidobacterium* and *Lactobacillus* genera were negatively correlated with phenol (R < −0.650, FDR < 0.008). Interestingly, the relative abundance of fecal *Bifidobacterium* was also negatively correlated with bloating and abdominal pain of patients. On the contrary, *Enterobacter* and *Serratia* were correlated negatively (R < −0.700, FDR < 0.02) with acetic acid, 2-methyl-propanoic acid, D-limonene, and 2-nonanal, and positively associated (R > 0.690, FDR < 0.03) with 1-butanol and phenol concentrations.

## 4. Discussion

In this study, we investigated common functional gastrointestinal symptoms, namely, abdominal pain, bloating, and bowel habits, in relation to intestinal permeability and fecal microbiota in patients treated with a specific probiotic containing *B. longum* BB536 and *L. rhamnosus* HN001 plus vitamin B6. In this cross-over, double blind, placebo-controlled, short-term study, we found consistent amelioration of some gastrointestinal symptoms, intestinal microbiota, and related metabolism with ZR, compared with placebo.

Also, as LI affects approximately 75% of the world population [34], we enrolled the cohort with an established diagnosis of lactose intolerance and persisting symptoms after lactose-free diet for at least six months. We cannot exclude that persisting symptoms could originate from other dairy compounds, namely fats and proteins [35]. Indeed, fats in some dairy products were more likely to cause symptoms in at least two studies [36,37]. In addition, we cannot exclude that beta-casein proteins also activate gastrointestinal µ-receptors, which, in turn, stimulate motility and abdominal symptoms [38,39]. Strain-specific bacterial metabolism might operate in the selection of probiotics and symbiotics to decrease the symptoms of gastrointestinal disorders, including LI [34]. Indeed, previous studies demonstrated that *B. longum*, as well as *L. rhamnosus*, improves general gastrointestinal symptoms, promotes lactose tolerance, and encourages a positive shift in gut microbial composition [40,41].

Bloating was most effectively corrected by ZR, according to the work of Zhu et al. [42]. Even with the lower intake of fibers than the recommended guidelines, constipation showed a trend towards improvement. Similar results have been previously described by Riezzo et al. [43]. Our data showed that the administration of *B. longum* BB536 and *L. rhamnosus* HN001 plus vitamin B6 for one month did not improve the intestinal permeability and abdominal pain. However, Yüce et al. [44] demonstrated that abdominal pain covers an important slice, but not the totality of LI patients with FGIDs. Recent research [45] suggests that probiotic bacteria formulation containing *L. acidophilus*, *L. rhamnosus*, *L. casei*, *Bifidobacterium breve*, *S. thermophilus*, *B. longum*, and *B. infantis* ameliorate flatulence and bloating in lactose malaborber patients. Our study suggests that similar results also stand for our patients.

Thirty days of treatment with *B. longum* BB536 and *L. rhamnosus* HN001 led to a positive shift in intestinal microbial composition. This modulation could be also the result of the intake of vitamin B6, according to the role of this vitamin in the metabolic pathways of microbes [46]. ZR drove an enrichment of genus *Bifidobacterium*. It is associated with lactose tolerance [47] and has been demonstrated to show relief in LI subjects with persistent FGIDs [41].

Recently, it was shown that short-chain galactooligosaccharide (GOS) induced a shift in the microbiome, increasing the relative amount of healthy promoting bacteria (*Lactobacillus*, *Bifidobacterium*, *Faecalibacterium*, and *Roseburia*), and a lowering of potentially detrimental species (*Enterobacteriaceae*) in LI patients [48]. In the present study, we found that the ratios of *Lactobacillus–Bifidobacterium* and *Bacteroides–Enterobacteriaceae* were respectively higher in ZR compared with those in PL. A lower ratio was demonstrated to be associated with food intolerance and allergy [49]. According to previous findings [48], *Bifidobacterium* showed a negative correlation with bloating and abdominal pain. Interestingly, the relative amount of some *Bacteroides* species was higher in PL compared with ZR. Some species of *Bacteroides* showed a pro-inflammatory effect [50]. In addition, a different balance of *Lactobacillus* species was found in ZR compared with PL. *Lactobacillus* species are known to be involved in LI relief [51]. However, other species have also been demonstrated to be involved in lactose digestion. Indeed, Phillips et al. [52] demonstrated that *Escherichia* spp. are involved in lactose breakdown.

The beneficial role in human gut health of acetic, propionic, and butanoic acids, against different types of disease, has already been well described [52,53]. Short chain fatty acids (SCFAs) might contribute to responses of immune cells in diseases associated with alterations in populations of commensal bacteria (dysbiosis) [54,55]. Fibers or dietary plant polysaccharides, such as non-digested oligosaccharides and proteins, are fermented by the microbiota populations to produce SCFAs [56]. In particular, *Bifidobacterium*, belonging to the phylum Actinobacteria and the groups of lactobacilli, are the main bacteria playing a central role in SCFAs’ metabolism [57]. The higher level of acetic acid and 2-methyl-propanoic acid during the treatment could be related to the increase of *Bifidobacteirum* [58]. Accordingly, the amount *Bifidobacterium* was positively correlated with acetic acid and 2-methyl-propanoic acid. Acetate is an important SCFA present in the colon, which could have a trophic effect on the colonic epithelium not only by local action, but also by raising the mucosal blood flux [59]. In contrast to SFCAs, the products of amino acid fermentation impair colon health, and so their presence is undesirable. *Bifidobacterium* and *Lactobacillus* genera negatively correlated with the concentration of phenol in fecal samples. Clostridiaceae are one of the main bacterial groups, which synthesizes some metabolic products (e.g., phenols, p-cresol, certain indole derivatives) that are potentially toxic for humans [30,60]. Positive correlations were found between the relative amount of Proteobacteria genera (*Serratia* and *Enterobacter*) and the concentration of phenol in fecal samples of LI patients. A significant decrease of fecal phenol observed in ZR compared with PL could be explained by considering the *Bifidobacterium* shift in gut microbial population.

## 5. Conclusions

According to our results, probiotic and vitamin B6 treatment may be useful to alleviate symptoms in subjects with LI and persistent FGIDs through a positive modulation of gut microbial composition and relative metabolism. One trial limitation is the relatively short duration of treatment. The randomized, cross-over design of the protocol, however, partly overcomes this problem, although longer studies need to be addressed in the near future.

## Figures and Tables

**Figure 1 nutrients-11-00886-f001:**
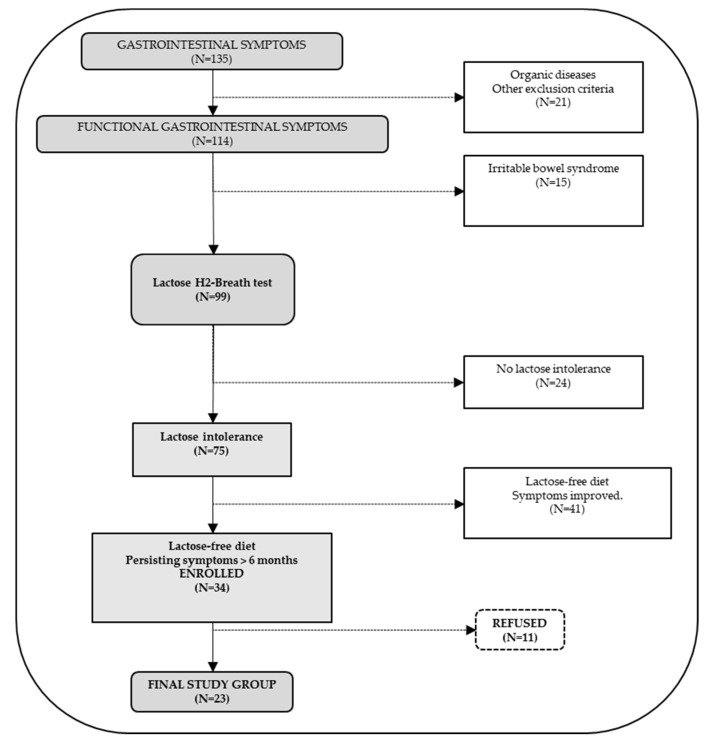
Consort flow-chart of screened patients. Starting from a general group of 135 symptomatic patients, and after exclusion of 21 patients with organic diseases, a subgroup of 114 patients had functional gastrointestinal symptoms. After further exclusions of 15 patients with Rome IV criteria, 99 patients underwent lactose H2 breath test and 75 were lactose intolerant, of which 34 had persisting symptoms following a lactose-free diet for at least six months, and were enrolled. Following the refusal of 11 subjects, the final study group consisted of 23 patients.

**Figure 2 nutrients-11-00886-f002:**
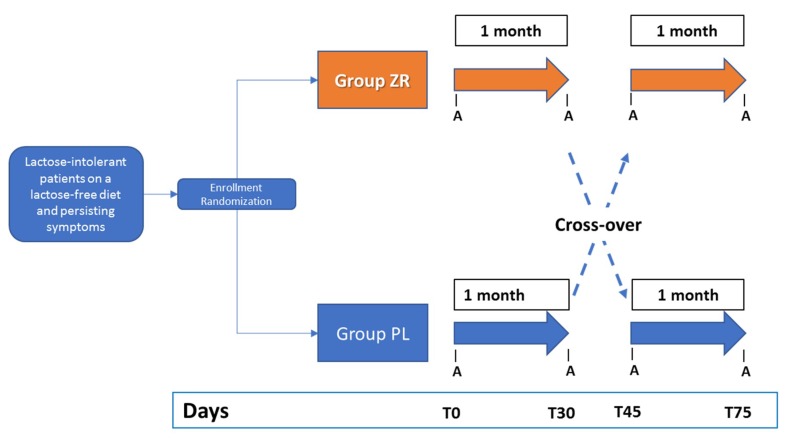
Crossover design of the study and timing of clinical evaluations. During the run-in (Ri) period, subjects fitting the inclusion criteria were randomized to receive either probiotic (ZR: 3 g as packets containing *Bifidobacterium longum* BB536 four billion CFU, *Lactobacillus rhamnosus* HN001 one billion CFU with B6 vitamin 1.4 mg) or placebo (PL: similar packets containing maltodextrins, corn starch, silicon dioxide, and no probiotic). The study included a first period of one month of treatment followed by 15 days of wash-out, and a second period of one month of treatment after cross-over. LI—lactose intolerant.

**Figure 3 nutrients-11-00886-f003:**
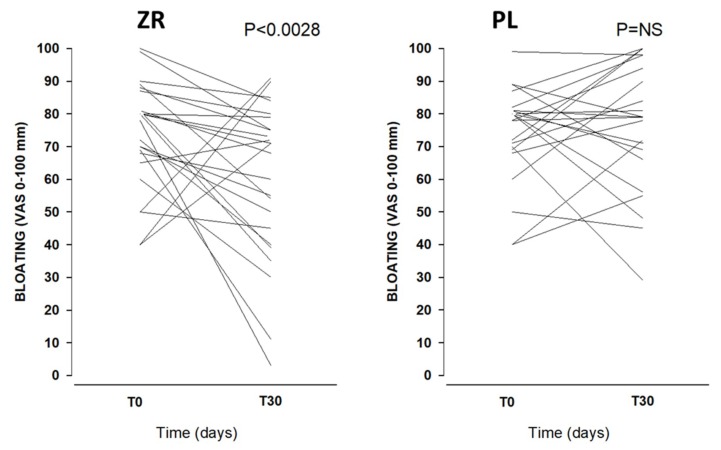
Representation of bloating (visual analogue scale (VAS) 0–100 mm) in 23 patients at baseline (T0) and after 30 days (T30) of treatment (ZR) and placebo (PL) by spaghetti graft.

**Figure 4 nutrients-11-00886-f004:**
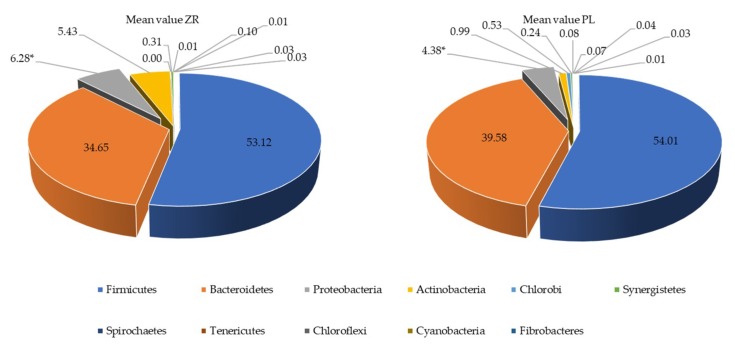
Relative proportions (percent) of phylum in the fecal samples of treated patients (ZR) and those of placebo (PL) patients.

**Figure 5 nutrients-11-00886-f005:**
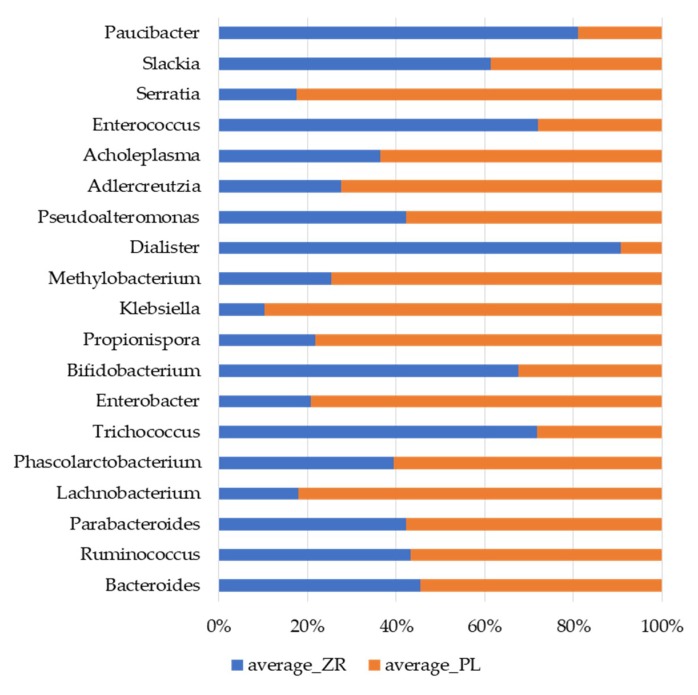
Relative proportions (percent) of genera showing significant (*p* < 0.05) differences between the fecal samples of treated patients (ZR) and those of placebo (PL) patients.

**Figure 6 nutrients-11-00886-f006:**
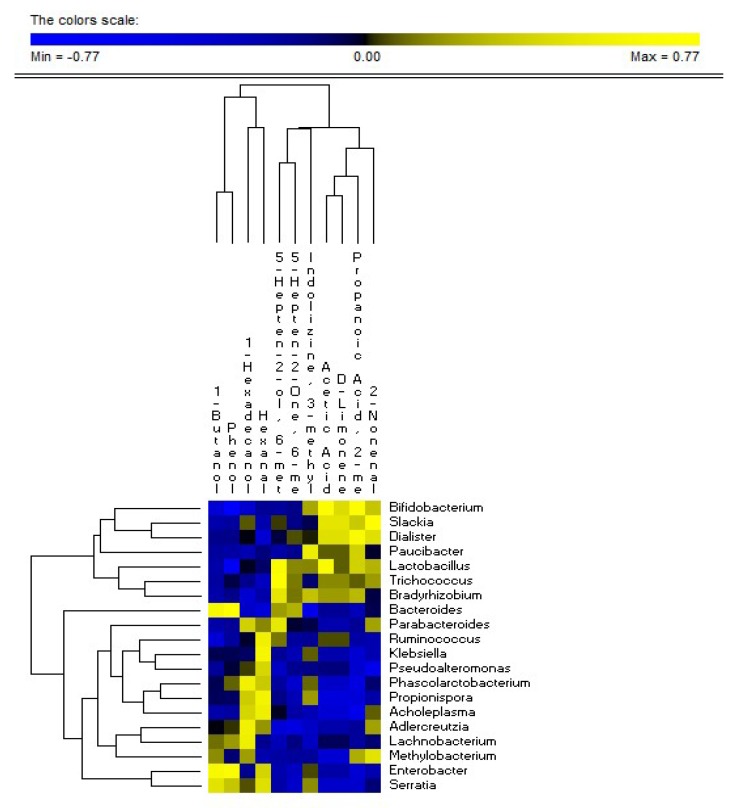
Significant correlations found between fecal microbiome and metabolome of lactose intolerant patients.

**Table 1 nutrients-11-00886-t001:** Baseline characteristics of the study patients (*N* = 23).

**Age, years**	48 ± 3.1 (48)
**Females, *n*. (%)**	19 (83%)
**BMI (Kg/m^2^)**	23.2 ± 0.68 (22.3)
**Symptoms**	
Bloating (VAS, mm)	69 ± 5.4 (80)
Abdominal pain (VAS, mm)	61 ± 4.6 (62)
**Bowel habits**	
Bristol Score (range 1–7)	3 ± 0.38 (2)

Legend: BMI, body mass index; VAS, visual analogue scale; data are mean ± SEM (median).

**Table 2 nutrients-11-00886-t002:** Volatile organic compounds (VOCs). Concentration (min, max) of VOCs (ppm) headspace fecal samples of treated (ZR) and placebo (PL) patients.

Compounds	ZR	PL	*p*-value
1-Butanol	0 (0, 0)	11.22 (0, 12.58)	0.037
5-Hepten-2-ol, 6-methyl-	51.74 (5.98, 72.3)	30.81 (0.79, 39.28)	0.029
Phenol	4.12 (0, 8.56)	19.78 (3.48, 28.56)	0.048
1-Hexadecanol	4.03 (0, 6.29)	8.17 (0, 13.94)	0.043
Acetic Acid	44.53 (26.62, 63.82)	23.34 (7.72, 37.47)	0.042
Propanoic Acid, 2-methyl-	29.78 (13.4, 45.04)	11.26 (0.71, 17.35)	0.009
Hexanal	5.81 (0, 13.22)	11.3 (2, 19.38)	0.009
2-Nonenal	5.38 (2.24, 8.9)	3.35 (0, 4.34)	0.045
Indolizine, 3-methyl-	248.96 (87.94, 372.62)	99.53 (31, 93.91)	0.045
5-Hepten-2-One, 6-methyl-	310.76 (55.31, 396.75)	124.03 (19.91, 210.84)	0.048
D-Limonene	87.43 (11.19, 187.9)	483.68 (55.46, 804.44)	0.042

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
