# Peer review of "Effects of Bifidobacterium longum and Lactobacillus rhamnosus on Gut Microbiota in Patients with Lactose Intolerance and Persisting Functional Gastrointestinal Symptoms: A Randomised, Double-Blind, Cross-Over Study"

_nutrients, 2019, doi:10.3390/nu11040886_

Round 1

Reviewer 1 Report

Vitellio et al describe a prospective randomized crossover clinical and fecal microbiome study on 23 patients, initially diagnosed with lactose intolerance, but who failed to improve symptomatically after a 6 month trial of lactose free diet. After this period, the 23 patients from the original group of LI patients were given placebo or 2 probiotics (B. longum BB536 and L rhamnosus) along with vitamin B6 (dose 1.4g/d which well approximates recommended daily requirements) for a period of 1 month, 2 week wash out period and then crossed over to receive the other product. Lactose free diet was continued until the end. The hydrogen breath test was used to classify patients into severity of symptoms which correlated with quantities of hydrogen production. Randomization is well described as well as the cross over design. Symptoms were evaluated by questionnaires at the beginning and end of each crossover period. The fecal bacteria of relevance were evaluated by different methods. In addition metabolic products of relevance and intestinal permeability were evaluated. The results showed significant reduction only of bloating but significant microbial and metabolic changes after the test products were given.

Comments: This is a sophisticated study of the effects of the probiotic combinations on clinical symptoms and fecal microbial alterations. As such it is generally well written and important for possible therapy of functional symptoms. However I have several questions.

The way the paper is written it is unclear what role lactose intolerance has to do with the target population. Part of the confusion may relate to equalizing the terms lactose intolerance to lactose malabsorption. Although the authors point out the difference between LI and LM on pg 2 (lines 49-52) they still refer to the target population as LI with FGID. However is not the definition of LI encompassing persons who consume lactose. The current test population was on a total of 8 months of lactose free diet. Therefore it is difficult to assign symptoms to lactose ingestion.

It should be noted that FGID which includes IBS  (symptoms very close to those of LI) occur with about similar frequency in both lactase non persistent and lactose persistent people (Barr, S.I. Perceived lactose intolerance in adult Canadians: A national survey. Appl. Physiol. Nutr. Metab.2013, 38, 830–835, Farup, P.; Monsbakken, K.; Vandvik, P. Lactose malabsorption in a population with irritable bowel syndrome. Scand. J. Gastroenterol. 2004, 39, 645–649 Yang, J.; Deng, Y.; Chu, H.; Cong, Y.; Zhao, J.; Pohl, D.; Fox, M. Prevalence and presentation of lactose intolerance and effects on dairy product intake in healthy subjects and patients with irritable bowel syndrome. Clin. Gastroenterol. Hepatol. 2013, 3, 262–268). As a result one could consider the 23 patients in the study as primarily having FGID and LI is irrelevant.

 The discussion however, also mentions the probiotics improved lactose tolerance, but how was this conclusion made. The methods do not talk about a breath hydrogen test after a repeat lactose challenge which would support the contention if less breath hydrogen and symptoms were produced.

In conjunction with the point of lactose free diet I think the authors should describe the diet a little further. Was dairy excluded, was dairy, that was lactose free used or were there dairy substitutes. This point may be relevant in light of other nutrient (not only lactose) in dairy which may provoke symptoms (eg fats, casein). Supplementary Table 1 gives only indications of micronutrients. For example more animal protein is noted but the source is unclear.

The rationale for adding Vitamin B6 and the mechanism of action in improving symptoms is not well described in the methods or the discussion. The reference given does not really clarify this point.

Table 2 is not clear; the letter “a” is used to denote a significant change at p < 0.05. The letter “b” presumably not significant is unnecessary.

Several contextual and spelling errors were noted. Pg 2, line 56 change relief, pg2 line 64 add “as”  after defined, Pg 7, line 258 change Female to Females, Pg 9, line 322. Change differed to “differ”, Pg 10, line 339, faces should be feces or faeces, Pg 11, line 348 should be relative (not relatively), Pg 12, line 376 remove “s” from malabsorbers or remove the word patients.

Author Response

Dear Reviewer,

We want to thank you and the reviewer 2 for the significant contributions in improving the paper “Effects of Bifidobacterium longum and Lactobacillus rhamnosus on gut microbiota in patients with lactose intolerance and persisting functional gastrointestinal symptoms: A randomised, Double-Blind, Cross-Over Study” by Paola Vitellio et al. We went through all points raised and below, you will find the point-by-point answer to each comment which is now marked in red in the text.

Best regards,

Maria De Angelis

Piero Portincasa

Reviewer 2 Report

Title: Accurate

Abstract:

What is the hypothesis?

What are the objectives?

Introduction:

53: The inadequate splitting of lactose in the colon is associated: The inadequate splitting is in the small bowel

82: The lactose hydrogen breath test (LH2-BT): why was the 25 gm of lactose given in milk?

119: with B6 vitamin 1.4 mg:  What is the logic for giving B6

120. placebo (PL: similar packets containing maltodextrins, corn 120 starch: Why include starch products?

145: Ethical reviews: Excellent

155. questionnaires: What is the quantitative reproducibility with these scales?

153. Primary outcome was LI severity measured by the lactose hydrogen breath test:

Was the H2 BT repeated at randomization?

Where are these results?

174 Culturable microbiota: put in an electronic supplement

220. Intestinal permeability: put in an electronic supplement

294: Diet: excellent

279: Clinical Scores: excellent

293: microbiota and Table 2: put in electronic supplement

304: microbiome: excellent

328: metabolome: outstanding

356: discussion: please integrate microbiome and metabolome results from Figure 7

Figures: Good

Table: Good

References: See above (47)

Supplements: not accessible from pdf.

Author Response

Dear Reviewer,

We want to thank you and the reviewer 1 for the significant contributions in improving the paper “Effects of Bifidobacterium longum and Lactobacillus rhamnosus on gut microbiota in patients with lactose intolerance and persisting functional gastrointestinal symptoms: A randomised, Double-Blind, Cross-Over Study” by Paola Vitellio et al. We went through all points raised and below, you will find the point-by-point answer to each comment which is now marked in red in the text.

Best regards,

Maria De Angelis

Piero Portincasa

Round 2

Reviewer 1 Report

The paper is more clear now. There are 2 minor language corrections

Pg 2 paragraph 3 "In addition there are  evidences" Chnge to "is evidence"

Pg 10 2nd paragraph of discussion, end of 3rd line change "other daily"  to "other dairy"

Author Response

Thank you for your comment. The spelling errors were corrected.

-       “there is evidence” was used instead of "there are  evidences" (Pg. 2)

-       "other dairy" was used instead of "other daily" (Pg. 10).
